# Evaluation of Polycaprolactone Applicability for Manufacturing High-Performance Cellulose Nanocrystal Cement Composites

**DOI:** 10.3390/polym15163358

**Published:** 2023-08-10

**Authors:** Hyungjoo Lee, Woosuk Kim

**Affiliations:** 1The Research Institute of Building and Construction Technology, Kumoh National Institute of Technology, Gumi 39177, Republic of Korea; lhj1102@kumoh.ac.kr; 2Department of Architectural Engineering, School of Architecture, Kumoh National Institute of Technology, Gumi 39177, Republic of Korea

**Keywords:** polycaprolactone, cellulose nanocrystal, optimum mixing ratio, high-performance cement composite, filler effect hydration product

## Abstract

This experimental study examined the aplication effect of polycaprolactone (PCL), an organic resin material with excellent elasticity and ductility, on improving the mechanical performance of cellulose nanocrystal (CNC) cement composites. PCL was compared according to its shape, and in the case of Granules, which is the basic shape, interfacial adhesion with cement was not achieved, so a dichloromethane (DCM) solution was used to dissolve and use the Granules form. As a method for bonding PCL to the CNC surface, the CNC surface was modified using 3-aminopropyltriethoxysilane (APTES), and surface silylation was confirmed through Fourier transform infrared spectroscopy (FT-IR) analysis. In order to evaluate the dispersibility according to the application of PCL to the modified CNC, particle size analysis (PSA) and zeta potential analysis were performed according to the PCL mixing ratio. Through the PSA and zeta potential values, the highest dispersion stability was shown at 1 vol.%, the cohesive force of CNC was low, and the dispersion stability was high according to the application of PCL. According to the results of the dispersion stability evaluation, the degree of hydration of the dissolved PCL 1 vol.%, CNC-only specimens, and plain specimens were analyzed. CNC acted as a water channel inside the cement to accelerate hydration in the non-hydrated area, resulting in an increased degree of hydration. However, the incorporation of PCL showed a low degree of hydration, and the analysis of strength characteristics also showed a decrease of approximately 27% compared with that of plain specimens. This was because the bonding with SiO_2_ was not smooth owing to the solvent, thus affecting internal hydration. In order to investigate the effect of the PCL shape, the compressive and flexural strength characteristics were compared using PCL powder as an additional parameter. The compressive strength and flexural strength were improved by about 54% and 26%, respectively, in the PCL powder 15 wt% specimen compared to the general specimen. Scanning electron microscopy (SEM) analysis confirmed that the filler effect, which made the microporous structure denser, affects the mechanical performance improvement.

## 1. Introduction

High-performance fiber-reinforced cementitious composites (HPFRCCs), as next-generation construction materials following the development of high-performance construction materials, are characterized by redistribution of the tensile stress to the matrix adjacent to the crack via the crosslinking phenomenon of fibers after the initial tensile crack occurs. They are evaluated as construction materials with pseudo-strain-hardening properties through the dispersion of many microcracks, and research is being conducted to satisfy the toughness and deformation capacity for application to repair/reinforcement and seismic members [1,2,3].

They can reinforce concrete more effectively than conventional fibers, and research is being conducted to utilize nanomaterials as reinforcing materials for cement composites. Nanomaterials exhibit higher electrical conductivity than conventional micro and larger materials to achieve equivalent performance in terms of electrical conductivity by forming thermal and electrical networks through the capillary pores in the cement matrix [4]. Typically, to improve the performance and characteristics of an ordinary Portland cement (OPC)-based binder, carbon nanotubes (CNTs) are used to make it lighter than steel (less than 1500 kg/m^3^), with a low mass density and high aspect ratio. They can share more load in the OPC matrix and effectively reinforce the cement composite by providing a significantly larger interfacial contact area [5].

Polycaprolactone (PCL) has excellent affinity with other polymers; therefore, research has been conducted on blends. It has excellent elasticity and ductility and is a biocompatible polymer that is widely used in biomedical applications, such as medical devices and drug delivery [6,7,8].

Nanocellulose is used as a reinforcing material in various composite materials because of its high specific strength and stiffness and is used as an eco-friendly and sustainable material because it can be obtained from other biological elements, such as various plants, algae, and tunicates [9,10,11,12].

Cellulose contains both crystalline and amorphous regions. Among them, the structure obtained by isolating only the crystalline region is called cellulose nanocrystal (CNC) [13]. CNC has a low density (1.6 g/cm^3^), high tensile strength (7.5–7.7 GPa), longitudinal elastic modulus of 110–220 GPa, and transverse elastic modulus of 10–50 GPa, showing similar or equivalent strength to carbon and steel fibers. CNC can be divided into CNF and nanocellulose and can be applied to industrial fields such as bio, cosmetics, paper, and filtration devices [14,15]. In addition, CNC has a large number of hydroxyl groups distributed on its surface, which has a very good reactivity with other molecules; therefore, the desired materials can be attached to the CNC surface [16]. As a method of replacing small molecules, toluene, in which phenyl isocyanate, alkenyl succinic anhydride, coupling agent, and catalyst are dissolved, can be added to replace hydroxyl groups on the surface with these molecules [17]. There are two methods: attaching the polymer to the surface using the polymer itself as a grafting agent and synthesizing the polymer chain directly from the surface by causing a polymer polymerization reaction at the hydroxyl group [18]. Research is being conducted on the development of nanocomposites through surface modification to obtain better physical properties by increasing interfacial adhesion using high CNC reactivity. Cellulose, silk, and nanoparticles were used as fillers to improve the thermal stability and mechanical properties of polylactic acid (PLA), an eco-friendly biodegradable plastic that has been developed as a material to replace existing petroleum-based plastics [19,20]. PCL, which has excellent mechanical properties such as tensile strength, elongation, and biodegradability, and CNC showed high tensile modulus, complex viscosity, and storage modulus of the nanocomposite through twin-screw extrusion, demonstrating the effectiveness of improving physical performance [21,22,23]. 

As such, most research on CNC/polymer synthesis showing excellent physical performance has been conducted in the fields of bio, medicine, and packaging, and various studies are being conducted to improve the physical performance of polymers using the high mechanical properties of nanocellulose [24].

Research in the field of construction was conducted by Cao et al. and Montes et al., who evaluated rheology and mechanical performance according to the cement type (Type I/II and Type V), CNC type, and mixing amount (vol.% compared to that of cement), Refs. [25,26] and Lee et al., who investigated the effect of improving the mechanical performance of cement composites using CNC alone, such as the evaluation of long-term durability such as freeze-thawing, salt damage resistance, and carbonation test of CNC cement composites [27]. In addition, manufacturing cement composites using PCL was studied. A study on PCL incorporation showed that it directly affected the toughness, integrity, and continuity of cement mortar and showed the effect of improving strength and ductility as the PCL content increased [28].

Certain studies on the application of CNC and PCL, with excellent physical performance, to the construction field have been conducted individually for each material, and research on the application of nanocellulose/PCL to improve the physical performance of cement composites is lacking. In addition, for application to repair/reinforce, seismic members of nanomaterials in the field of cement concrete are based on carbon fiber.

Therefore, in this study, the following method was conducted to develop an original technology for manufacturing high-performance cement composites using CNC/PCL nanocomposites, and the effectiveness of PCL for manufacturing high-performance CNC cement composites was evaluated.

CNC surface modification for PCL graftingEvaluation of dispersion stability of CNC suspension mixed with PCLHydration product analysis through thermal analysis evaluationEvaluation of strength characteristics according to the PCL shape and mixing ratioMicrostructure analysis of specimen

## 2. Materials and Methods

### 2.1. Material Preparation

#### 2.1.1. Preparation of PCL 

The CNCs used in this study were purchased from CelluForce, Canada, and PCL was purchased from Daejeong Chemical and Frontier Chemics Co., Ltd., Seoul, Republic of Korea. Granules (CAPA 6500, MW: 50,000) and powder (CAPA 6506, MW: 50,000) shapes were used to investigate the degree of CNC attachment according to the shape and strength effect of cement composite manufacturing (Figure 1a,b). Table 1 shows the physical properties according to the PCL shape.

In the study by Lu et al. (2022), granular PCL was used after being heated and melted using a water bath heater because it did not form an interfacial bond with the cement composite due to its surface non-fluidity characteristics. However, the particles were large, and the melting time was long (approximately 24 h) [28]. Therefore, in this study, a dichloromethane (DCM) solution was used as the solvent after dissolution and mixed with CNC using an agitator (Figure 1c). In addition, the powder form was used without additional treatment.

#### 2.1.2. Preparation of CNC Powder

The raw material of CNC used in this study was powder that had a size of 1–50 μm in the form of a round (Figure 2a). The physical properties of the Cellulose Nanocrystals (CNC) were provided by CelluForce (Montreal, QC, Canada) and are shown in Table 2, and the amount of CNC used was 0.8 vol.% (compared to cement), which applied the optimal mixing conditions for CNC cement composites derived by Lee et al. [27]. These micro-sized CNC particles were placed in distilled water that was first purified for nano-sized use, and the agglomerated CNCs were first dispersed for 20 min using a magnetic stirrer. The distributed CNC showed a rod shape of 40–100 nm or less (Figure 2b). Subsequently, to prevent damage to the CNC owing to the heat generated during long-term ultrasonic dispersion, the beaker was placed in an acrylic water bath filled with ice, and ultrasonic dispersion was performed with an energy of 5000 J/g for 10 min.

#### 2.1.3. CNC-PCL Suspension

As a CNC surface modification method for synthesis with PCL, 0.5 g of CNC was put into a beaker containing 40 mL of ethanol and 10 mL of deionized water, and ultrasonic treatment was performed for 30 min for uniform dispersion. After the sonication was completed, 3.5 mL of APTES was added to the mixed solution, and the reaction was induced by stirring at room temperature for 6 h. Then, the solute was separated from the solvent using a centrifugal separator (FRONTIER 5000, FC5707, Gumi, Kumoh National Institute of Technology, Republic of Korea) at 4000 rpm for 10 min. The modified CNC was mixed with dissolved PCL and magnetic stirred for 20 min to prepare a CNC-PCL suspension.

### 2.2. CNC Surface Modification

Nanocellulose can be dispersed in strongly polar solvents owing to the strong interactions between surface hydroxyl groups and solvent molecules. However, hydrogen bonding between the nanofibers leads to aggregation at the micro level. In addition, because of its hydrophilic nature, nanocellulose is difficult to disperse in hydrophobic media and most polymer matrices when preparing composites [29]. Surface modification, such as changing the surface hydrophilicity, is an effective way to improve the dispersibility of nanocellulose [30,31].

The chemical functionalization of CNCs has been performed primarily to introduce stable negative or positive charges on the surface to obtain better dispersion and to tune the surface energy properties to improve compatibility, particularly when used with non-polar or hydrophobic materials [32,33].

Surface modification is required for good dispersibility of CNC and synthesis with other polymers. As a modification method, a method of modifying the surface of CNC through silylation, that is, silane grafting, has been demonstrated [34,35,36]. The silane used for CNC modification has different functional groups at both ends, such that one end can interact with an OH group, and the other end can interact with a functional group in the matrix to form a bridge between them [37]. As a typical silylating agent used for silylation, 3-aminopropyltriethoxysilane (APTES), which has a simple structure and is inexpensive, has been used [38], and a dimethylformamide (DMF) solution has been used to increase compatibility with polymers. This can improve the mechanical performance of nanocomposites [39,40].

### 2.3. Fourier Transform Infrared Spectroscopy (FT-IR)

In this study, the CNC surface was modified using APTES, and the modified CNC was extracted through centrifugation after adding APTES to the dispersed CNC suspension and stirring for approximately 3 h. To determine whether the extracted CNC was modified, FT-IR analysis was performed using the INVENIO X and Hyperion 2000 models (Gumi, Kumoh National Institute of Technology, Republic of Korea) and the spectral changes before and after modification were compared with the wave number in the analysis range of 600–4500 cm^−1^.

### 2.4. Particle Size Analysis (PSA) and Zeta Potential

PCL dissolved in the modified CNC suspension was prepared by stirring for approximately 3 h. PSA and zeta potential measurements were conducted to determine the effect of PCL on the particle size distribution and dispersion stability over time and analyzed according to ASTM D4187-82 [41] (Table 3). The PSA measured the particle size distribution for approximately 1 min and confirmed the size distribution according to the variable. In order to evaluate the dispersion stability and retention of CNC according to the contents (0.1, 0.3, 0.5, and 1 vol.% compared with those of CNC) after dissolving Granules-form PCL, measurements were taken after 7 days and 30 days.

### 2.5. Thermogravimetric Analysis (TGA)

In order to measure the degree of hydration of the CNC/PCL-incorporated cured specimens, TG-DTG analysis was performed using an Auto-TGA Q500 model in accordance with ASTM E 1131 [42]. A test specimen mixed with PCL (1 vol.%)/CNC (0.8 vol.%) suspension, specimen mixed with only CNC without PCL, and plain specimen without any additives were compared. The change owing to the heat of the sample was measured using a temperature–weight change curve with a temperature increase of 20 °C/min from 0 to 1000 °C.

### 2.6. Mechanical Properties

In order to determine the physical properties of the CNC/PCL-applied cement composite, a strength test was conducted using a universal testing machine (UTM). The compressive and flexural strengths were compared based on the mixing ratio of the dissolved granules and powders, including the control specimens.

### 2.7. Scanning Electron Microscopy (SEM) and Energy Dispersive X-ray Spectrometry (EDS)

SEM and EDS were conducted using the MAIA 3 LM model to analyze the internal shape and chemical components of the microstructure of the CNC/PCL cement composites. For an accurate observation, the inside of the equipment was created in a vacuum state for approximately 1 min before the start of the experiment.

### 2.8. Mixture Design

Table 4 lists the mixture design of the cement composite according to the shape and mixing of PCL. Because the moisture adsorption of CNCs with hydrophilic characteristics affects their surface behavior [43], a high-performance water-reducing agent was used in consideration of fluidity. In addition, 1 vol.% of PCL (Granules) dissolved through dispersion stability evaluation, and TGA analysis was used. In the case of powder, an experiment was conducted using 5, 10, and 15 wt.% of cement without solvent treatment to examine the effect on the strength of the cement composite.

## 3. Test Results

### 3.1. CNC Surface Silylation

As for the peak position according to the accumulated scans of FT-IR, some peaks in the spectrum of the CNC that underwent the surface modification process using APTES showed a change (Figure 3). The peak assignments for the peak positions are listed in Table 5 [44]. In the wavenumber region of 1580–1590 cm^−1^, a peak that was not present in the CNC before modification was generated; this corresponded to the N–H bending of the amine group by the introduction of APTES [45]. The absorption peak located at 3400 cm^−1^ showed –OH vibration [46]. The CH_2_ bending peak of the propyl group of APTES and the Si-O stretching peak of the silanol group appeared around 1460 cm^−1^ and 1140 cm^−1^, respectively. This confirmed that APTES was modified on the CNC surface. These new peaks indicated that the CNCs were successfully silylated.

### 3.2. Dispersibility

The size distribution of each variable in the PSA experiments is presented in Table 6. After 7 days of sample production, the overall distribution was approximately 80–100 nm, and after 30 days, it was approximately 60–80 nm, showing similar or reduced distribution. The average particle size decreased by approximately 10% at 0.5 vol.% after 7 days and by approximately 8% at 1 vol.% after 30 days (Figure 4a).

Notably, samples improved to moderate stability after 7 days and higher good stability after 30 days.

The dispersion stability of the sample containing PCL was higher than that of the sample not containing it. Even after the passage of time, no cohesive force was observed owing to the incorporation of PCL, confirming the dispersion stability of the CNC/PCL suspension (Figure 4b). The particle size distribution over time showed the highest stability at 1 vol.% with a distribution of 63–72 nm after 30 days.

### 3.3. Hydration Product Analysis

The internal microstructural mechanisms of CNC-bonded cement composites are illustrated in Figure 5. As shown in Figure 5, the incorporation of CNC promoted hydration through the role of a water channel in the non-hydrated area inside the cement, increasing the number of hydration products and improving the microstructure [24,47]. Based on these results, the effect of dissolved PCL (1 vol.%) incorporation with the highest dispersion stability on the hydration product was reviewed through the aforementioned dispersibility evaluation. To accurately confirm the result of reduced hydration products owing to the thermal change in the PCL cement composite grafted in the modified CNC, a DTG curve that differentiated the TGA test result is shown (Figure 6).

The TGA results were calculated using Bhatty’s cement hydration formula (Table 7). The formula consists of the amount of mass loss according to Equation (1) [48], where *W_dh_* (105–400 °C), *W_dx_* (400–500 °C), and *W_dc_* (500–900 °C) denote mass losses owing to dehydration, dehydrogenation, and decarbonization, respectively. A factor of 0.41 is used to convert the mass loss owing to decarbonization to the equivalent molecular weight of water [49]. Hydration degree α can be calculated using Equation (2), and a factor of 0.24 corresponds to the theoretical maximum number required for the complete hydration of cement [50].
(1)$$W_{b}=W_{dh}+W_{dx}+0.41(W_{dc})$$
(2)$$\alpha \left(\%\right)=\frac{W_{b}}{0.24}\times 100$$

The mass change in the hydration product decreased in a similar pattern because of the inherent hydrate decomposition at a constant temperature upon the application of heat. In the range of 105–400 °C, the dehydration reaction of C-S-H, which is an amorphous hydrate, occurred, and at 400–500 °C, chemical bond water was released. Ca(OH)_2_ lost its alkalinity through thermal decomposition. In the case of CaCO_3_, the weight was reduced by decomposing into CaO and CO_2_ at 500–900 °C. The main hydration products were C-S-H, Ca(OH)_2_, and CaCO_3_. The degree of hydration was 20.67% when only CNC was incorporated, which increased by approximately 29% compared with that of the PCL/CNC specimens. Consequently, it was judged that there is no effect on the hydration product inside the CNC cement composite according to the PCL content.

### 3.4. Strength Test

The 28-day strengths of the CNC cement composites and plain cement were compared according to the presence of PCL(Table 8), and the compressive strength tended to increase as the content of PCL in the form of powder increased (Figure 7a).

The CNC test specimen showed an average strength value of 45 MPa, showing an improvement of approximately 22% compared with 31 MPa of the plain specimen. The C-P-15 specimen showed a strength value of 51 MPa, with an improvement of approximately 54%, showing the highest improvement effect. This was judged to have an effect on the strength owing to the increase in hydration products because of the inclusion of CNC and the densification of the internal voids through the PCL. When the Granules was mixed, the strength value decreased by approximately 27% compared with that of the plain specimen, and a brittle fracture tendency was observed in all specimens.

In the case of the flexural strength, the C-P-10 and C-P-15 specimens improved by up to about 26% compared to the general specimens, but the ductility effect due to the PCL incorporation was not observed and showed a tendency to brittle fracture (Figure 7b). In addition, the decrease in strength of the Granules specimen (C-P-1) dissolved using DCM solution showed polarity as the ring structure of PCL was broken by catalyst or heat, and SiO_2_, one of the cement components, was a polar material. Therefore, PCL exhibited polarity, and the strength was evaluated to be improved through bonding (Figure 8) [51]. However, the bonding with SiO_2_ was not smooth owing to the influence of the solvent and had no effect on the micropore structure, unlike the powder form.

### 3.5. Shape and Chemical Composition of Microstructure

The internal microstructural shapes of the PCL/CNC cement composite and plain specimens are shown in Figure 9. The composition ratios, including Ca, Si, Al, and O, of the PCL/CNC specimens obtained through EDS analysis are shown in Table 9. In the case of the plain specimen, some internal voids occurred because of non-hydrated particles (Figure 9a). In the CNC and C-P-15 test specimens, the hydrated phases were covered by the C-S-H phase (red point), and the microstructure appeared to be more precise. This was because Ca(OH)_2_ reacted with soluble components such as SiO_2_ and Al_2_O_3_ inside the cement to form C-S-H or C-A-H. C-S-H and plate-shaped Ca(OH)_2_ (blue point) were observed, and needle-shaped CNC and ettringite were formed. Although the shapes of ettringite and CNC were similar, CNC exhibited a short and compact crystal form (Figure 9b). As shown in Figure 9c), the PCL particles were grafted in the CNC. According to the aforementioned TGA and strength test results, the hydration products slightly increased, and It was judged that the filler effect filling the internal voids had an effect on the improvement in mechanical performance.

## 4. Conclusions and Discussion

The experimental study examined the effects of PCL through physical and chemical analysis for the development of high-performance CNC cement composites using PCL, which has excellent affinity with other polymers, and the conclusions are as follows.

The surface-modified CNC suspension was observed to have a low cohesive force and increased dispersion stability through the incorporation of PCL, and this was determined to affect the inside of the CNC cement composite. However, PCL in the Granules form does not form an interfacial bond with cement; therefore, it must be used after melting by applying heat for a long time. Even if a solvent is used, using it as a cement composite is difficult because its bonding strength with cement is low.TGA and SEM analysis revealed that PCL in the form of powder was an organic resin material and had no effect on the hydration reaction of cement by itself. The effectiveness of PCL was confirmed by demonstrating that it played a role in improving the strength characteristics as a filler effect. In addition, as the mixing ratio increased, the strength value improved. The compressive strength improved by approximately 54%, and the flexural strength by approximately 25% in the C-P-15 specimen, showing the highest value. However, when CNC and PCL/CNC were used, the flexural strength value increased, but the irregular distribution inside the cement and the ductility effect did not show a clear behavior; therefore, this was judged to be less useful as a structural material.

## Figures and Tables

**Figure 1 polymers-15-03358-f001:**
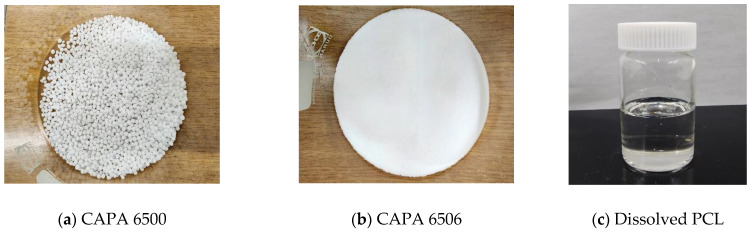
Shape according to PCL form.

**Figure 2 polymers-15-03358-f002:**
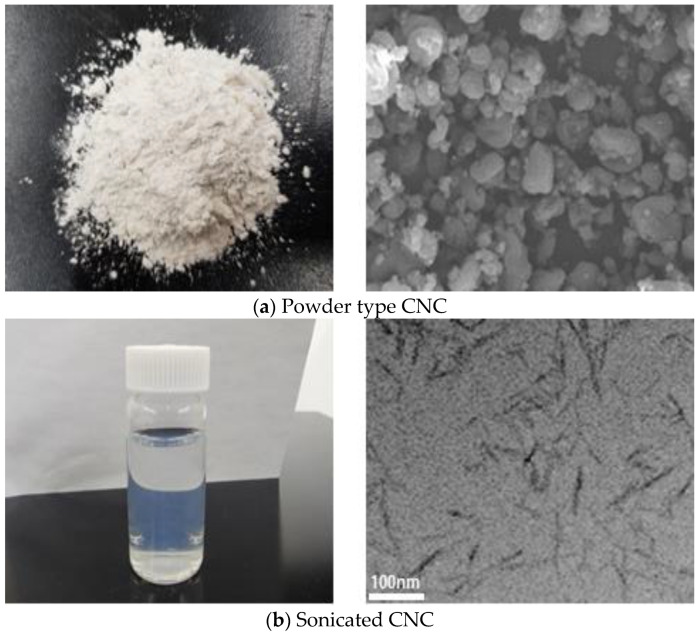
CNC shape before and after dispersion.

**Figure 3 polymers-15-03358-f003:**
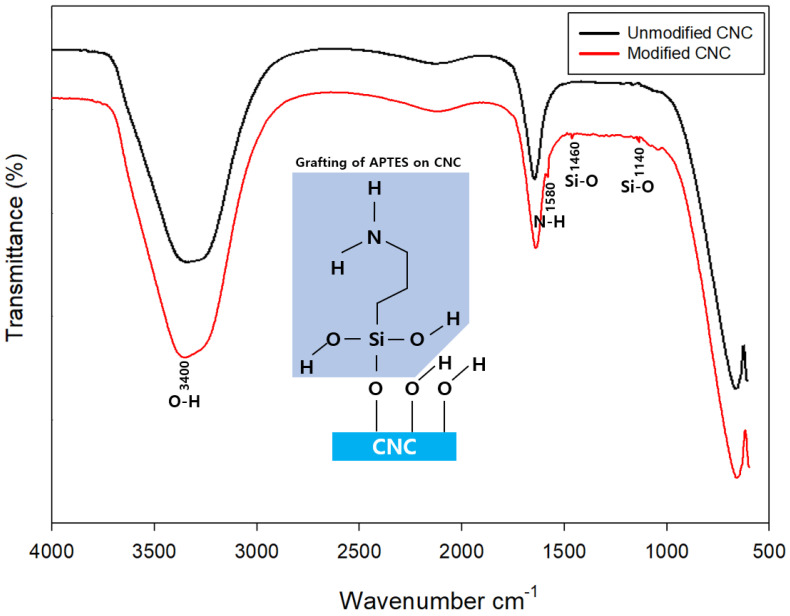
FTIR spectra of CNC surface modified.

**Figure 4 polymers-15-03358-f004:**
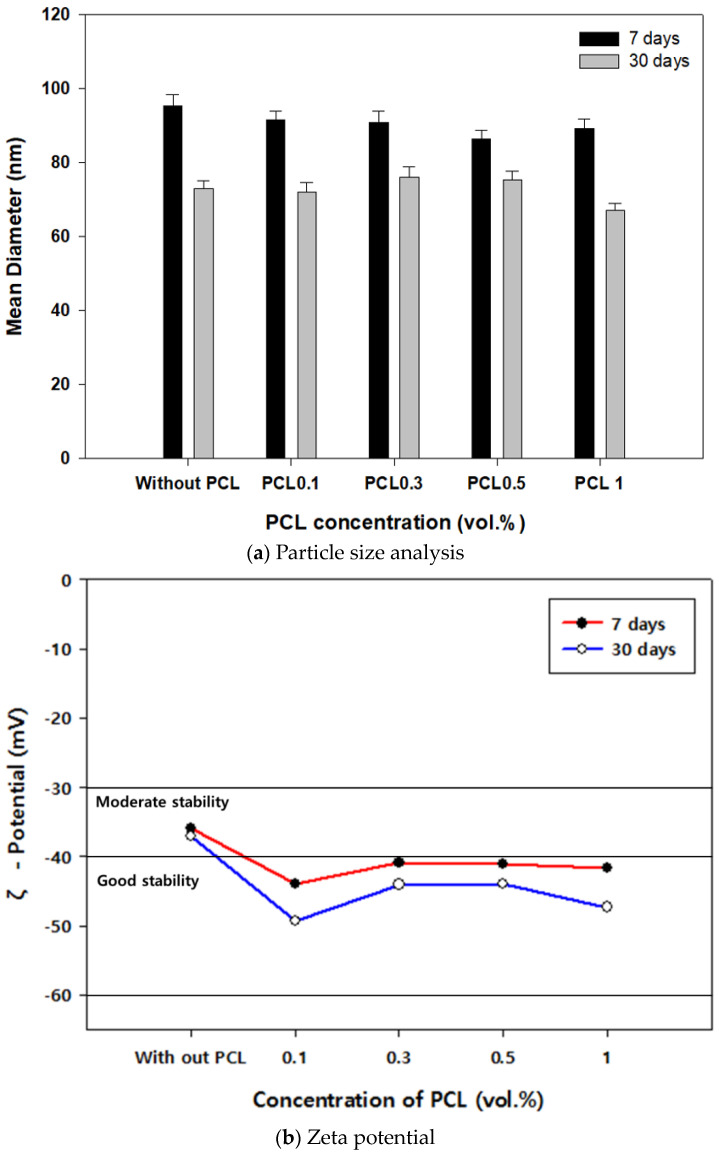
CNC/PCL dispersibility test results.

**Figure 5 polymers-15-03358-f005:**
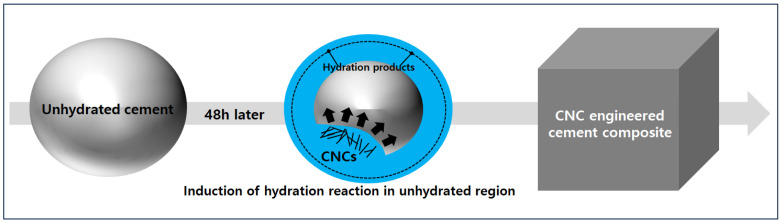
Schematic of the microstructure of CNC cement composites [27,47].

**Figure 6 polymers-15-03358-f006:**
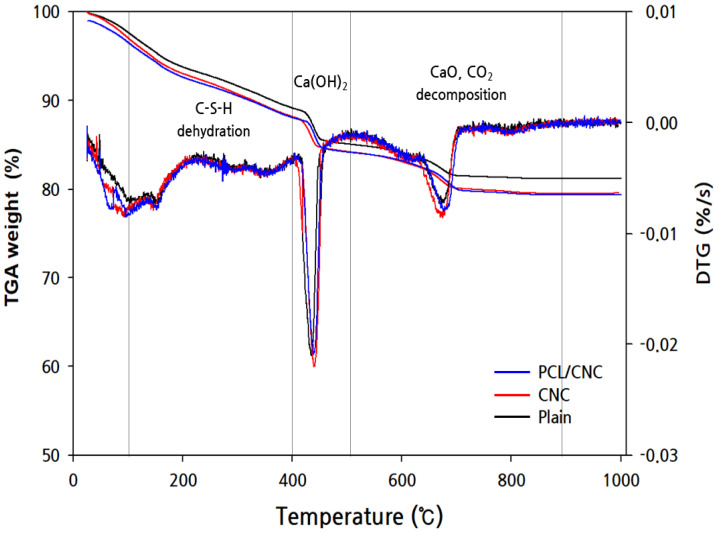
TG-DTG curve of hydration product reduction according to heat change.

**Figure 7 polymers-15-03358-f007:**
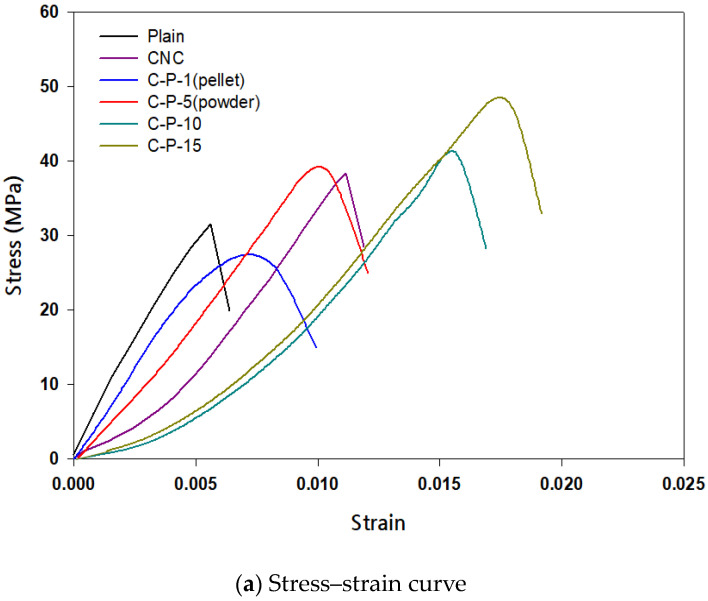
Strength test results according to parameters.

**Figure 8 polymers-15-03358-f008:**
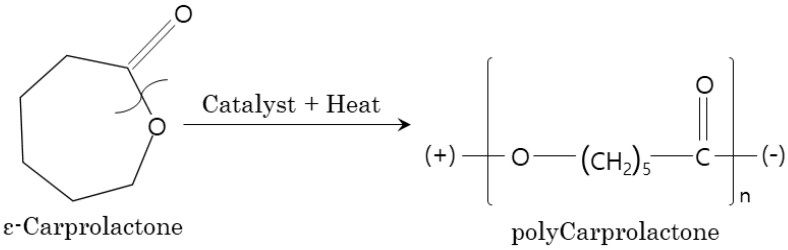
Changes in the PCL structure owing to heat.

**Figure 9 polymers-15-03358-f009:**
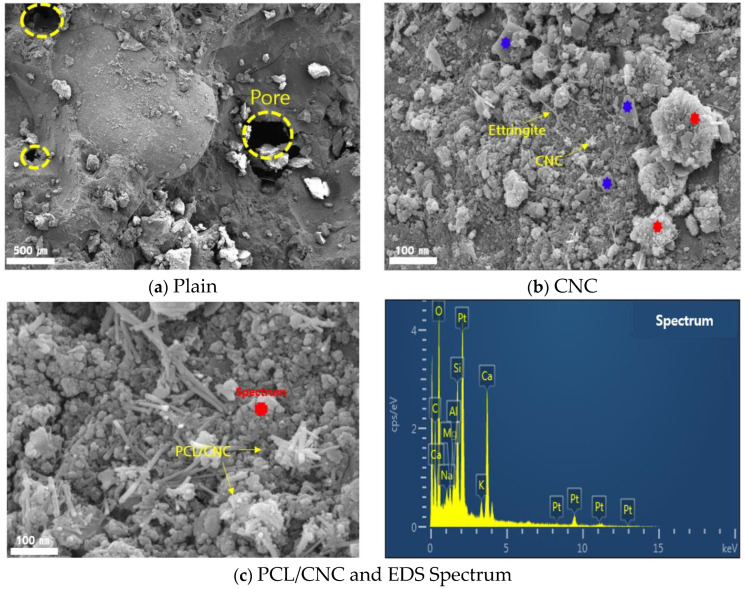
SEM and EDS analysis according to variables.

**Table 1 polymers-15-03358-t001:** Physical properties according to PCL shape.

Classification	Form	Molecular Weight	Color	Melting Point (°C)	Density (g/cm^3^)	Chemical Formula
CAPA 6500	Granules	50,000	White	58–60	1.1	(C_6_H_10_O_2_)_n_
CAPA 6506	Powder	50,000	White	58–60	1.1	

**Table 2 polymers-15-03358-t002:** Physical properties of CNCs used.

Form	Color	Crystallite Density	Particle Diameter	Particle Length	pH
Powder	White	1.5 g/cm^3^	2.3–4.5 nm (by AFM)	44–108 nm (by AFM)	6–7

AFM: Atomic Force Microscope.

**Table 3 polymers-15-03358-t003:** Index of evaluation of dispersion stability according to the zeta potential value [41].

Zeta Potential, ζ (mV)	Stability Behavior of the Colloid
From 0 to ±5	Rapid coagulation or flocculation
From 10 to ±30	Incipient instability
From 30 to ±40	Moderate stability
From 40 to ±60	Good stability
More than ±61	Excellent stability

**Table 4 polymers-15-03358-t004:** CNC/PCL cement composite mixing design.

Classification	W/C	C/S	CNC (vol.%)	PCL		APTES (wt.%)	S.P (vol.%)
				Granules (vol.%)	Powder (wt.%)		
Plain	1:2	1:3	-	-	-	-	-
CNC			0.8	-	-	-	1
C-P 1				1	-	3	
C-P 5				-	5		
C-P-10				-	10		
C-P-15				-	15		

**Table 5 polymers-15-03358-t005:** FT-IR spectrum table [44].

Peak Position (cm^−1^)	Peak Assignment
1000–1250	Si-O stretching of Si-O-Si crosslinked
1300–1400	CH_2_ and CH_3_ scissoring
1600–1670	C=C-H axial deformation
3200–3700	Axial deformation of Si-OH group OH

**Table 6 polymers-15-03358-t006:** Size range according to variable (unit: nm).

Classification	Without PCL	PCL Concentration (vol.%)			
		0.1	0.3	0.5	1
7 days (Standard deviation)	89–103 (3.07)	86–95 (2.22)	83–95 (3.22)	80–90 (2.37)	83–95 (2.51)
30 days (Standard deviation)	68–97 (2.12)	68–88 (2.52)	68–88 (2.58)	69–90 (2.34)	63–72 (1.97)

**Table 7 polymers-15-03358-t007:** Comparison of the degree of hydration using Bhatty’s method.

Tempering (°C)	Mass Loss (%)		
	Plain	CNC	PCL/CNC
105–400 ($W_{dh}$)	2.36	2.96	2.28
400–500 ($W_{dx}$)	0.81	1.24	0.72
500–900 ($W_{dc}$)	1.12	2.89	1.08
$W_{b}$	3.62	5.38	3.44
$DoH,\alpha \left(\%\right)$	17.01	20.67	16.01

**Table 8 polymers-15-03358-t008:** Strength test results (unit: MPa).

	Specimen	Plain	CNC	C-P-1 (Pellet)	C-P-5 (Powder)	C-P-10	C-P-15
Mean Strength							
Compressive		31.5	41.2	28.5	42.3	44.1	48.8
Flexural		3.5	4.2	2.9	4.2	4.7	4.5

**Table 9 polymers-15-03358-t009:** PCL/CNC specimen EDS results.

Chemical Compositions (wt%)					
C	O	Al	Si	Ca	Others
21.2	50.7	2	6.9	15	4.2

## Data Availability

The data presented in this study are available on request from the corresponding author.

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
