# Peer review of "Evaluation of Polycaprolactone Applicability for Manufacturing High-Performance Cellulose Nanocrystal Cement Composites"

_polymers, 2023, doi:10.3390/polym15163358_

Round 1

Reviewer 1 Report

Abstract:

- The abstract is poorly written and contains many badly constructed sentences. For example: "Hydration products increased by increasing the degree of hydration in the unhydrated area".

- The use of acronyms (DCM) in abstract should be avoided, unless they are defined.

Methodology:

- There are confusing sentences. For example:

Line 125: The sentence "In the study by Lu et al. (2022), when PCL in the form of a Granules was used, it was 125 heated and melted using a water bath heater because it did not form an interfacial bond 126 with the cement composite." is confusing.

Line 134-The raw material of CNC used in this study was powder that had a size of 1–50 ㎛ 134 in the form of a round membrane. What is the meaning of "round membrane"? 

- Line 145 - Table 2: How the physical properties of the CNC are examined?

- Line 169: What is the mode of FTIR analysis? ATR or transmittance?

- Line 176: pH of the suspension for Zeta potential measurement should be provided.

- Line 185: Any nitrogen gas purging for the TGA analysis?

- Line 193: Details on how the cement board specimen were produced should be clearly explained.

In general, the methodologies are too lengthy and out-of-focus. There are literature review (with citation) in the description of experimental procedure, making it hard to understand the procedures. The authors also did not clearly explain how the PCL-CNC granules are prepared. 

Results and discussion

Which PCL composition was used for preparing the cement composition? Why the composition was chosen? This information is not provided in the manuscript. 

Figure 4(a): Basically this figure is meaningless. It just shows raw data obtained by the instrument. Suggest to just report the average diameter.

Figure 6: The figure caption should be improved.

Line 266; Table 6: The weight reduction of the samples should also involve dehydration, decarbonization, decomposition, etc. Therefore the discussion should be from the angle of CNC and the cement fraction. 

Table 7 should be placed at Methodology section.

Figure 7: Instead of showing the raw data in stress vs strain curve, the results should be calculated and tabulated to support the discussion.

Conclusion

The conclusion is too lengthy. It should be focused on the major findings and contribution of the study.

Recommendation

The manuscript requires major revision before it can be considered for publication. 

Author Response

Thank you very much for your comments. The authors will answer you as follows. Please see the attachment.

Abstract:

- The abstract is poorly written and contains many badly constructed sentences. For example: "Hydration products increased by increasing the degree of hydration in the unhydrated area".

- The use of acronyms (DCM) in abstract should be avoided, unless they are defined.

A: The Abstract has been revised.

Methodology:

- There are confusing sentences. For example:

- Line 125: The sentence "In the study by Lu et al. (2022), when PCL in the form of a Granules was used, it was 125 heated and melted using a water bath heater because it did not form an interfacial bond 126 with the cement composite." is confusing.

A: The contents have been revised.

In the study by Lu et al. (2022), When using granular PCL, it was used after being heated and melted using a water bath heater because it did not form an interfacial bond with the cement composite due to its surface non-fluidity characteristics.

- Line 134-The raw material of CNC used in this study was powder that had a size of 1–50 ãŽ› 134 in the form of a round membrane. What is the meaning of "round membrane"? 

A: The contents have been revised.

The raw material of CNC used in this study was powder that had a size of 1–50 ㎛ in the form of a round (Fig. 2(a)).

- Line 145 - Table 2: How the physical properties of the CNC are examined?

A: The contents have been revised.

The physical properties of the Cellulose Nanocrystals (CNC) were provided by CelluForce and are shown in Table 2.,

- Line 169: What is the mode of FTIR analysis? ATR or transmittance?

A: It is in Transmittance mode and the graph has been revised.

- Line 176: pH of the suspension for Zeta potential measurement should be provided.

A: No pH data were obtained in this experiment. Nevertheless, as indicated in Table 5, the experimental results for dispersion stability were written using the zeta potential (mV) value in accordance with ASTM D4187-82. Thank you for your understanding.

- Line 185: Any nitrogen gas purging for the TGA analysis?

A: Nitrogen gas purging was employed to prevent any reaction with the sample and to eliminate the influence of chemical reactions, ensuring that the experimental results were not distorted or compromised.

- Line 193: Details on how the cement board specimen were produced should be clearly explained.

In general, the methodologies are too lengthy and out-of-focus. There are literature review (with citation) in the description of experimental procedure, making it hard to understand the procedures. The authors also did not clearly explain how the PCL-CNC granules are prepared. 

A: Existing literature used for experimental procedures describes the basis for the experiments used in this study. Thank you for your understanding. And the contents of PCL-CNC suspension was added in the manuscript.

2.1.3 CNC-PCL suspension

As a CNC surface modification method for synthesis with PCL, 0.5 g of CNC was put into a beaker containing 40 mL of ethanol and 10 mL of deionized water, and ultrasonic treatment was performed for 30 minutes for uniform dispersion. After the sonication was completed, 3.5 mL of APTES was added to the mixed solution, and the reaction was induced by stirring at room temperature for 6 hours. Then, the solute was separated from the solvent using a centrifugal separator (FRONTIER 5000, FC5707) at 4000 rpm for 10 minutes. The modified CNC was mixed with dissolved PCL and magnetic stirred for 20 minutes to prepare a CNC-PCL suspension

Results and discussion

Which PCL composition was used for preparing the cement composition? Why the composition was chosen? This information is not provided in the manuscript. 

A: Added chemical formula to Table 1.

Figure 4(a): Basically this figure is meaningless. It just shows raw data obtained by the instrument. Suggest to just report the average diameter.

A: The graph in Figure 4(a) has been revised to mean data values.

Figure 6: The figure caption should be improved.

A: Figure 6 caption has been revised.

Line 266; Table 6: The weight reduction of the samples should also involve dehydration, decarbonization, decomposition, etc. Therefore the discussion should be from the angle of CNC and the cement fraction. 

Table 6 uses Bhatty's cement hydration formula and shows the mass loss due to dehydration, dehydrogenation, and decarbonization, respectively, for Wdh, Wdx, and Wdc. In addition, the mixture design used in the test was revised.

Table 7 should be placed at Methodology section.

A: Table 7 has been revised to the methodology section.

Figure 7: Instead of showing the raw data in stress vs strain curve, the results should be calculated and tabulated to support the discussion.

A: The strength test result values are tabulated and shown.

Conclusion

The conclusion is too lengthy. It should be focused on the major findings and contribution of the study.

A: Future research contents were deleted and only the contents of this research result were displayed.

Author Response

Thank you very much for your comments. The authors will answer you as follows.  Please see the attachment.

  1. The abstract needs to be precise, with the basic ‘why, how and what’ briefly described.

A: The manuscript has been revised

  1. There are irregular text/phrases throughout the manuscript, for example

Page 1, Line 33

‘…filler effect that made the micropore structure denser, affected the mechanical performance

improvement.’

A: The manuscript has been revised

filler effect, that made the microporous structure denser, affects the mechanical performance improvement.

Page2, line 45

‘..for application to repair/reinforcement and seismic members.’

A: The manuscript has been revised

In addition, for application to repair/reinforcement and seismic members of nanomaterials in the field of cement concrete are based on carbon.

Page 2, line 49 - repeated phrase

‘Nanomaterials exhibit higher electrical conductivity than conventional micro and larger

materials to achieve equivalent performance in terms of electrical conductivity by forming,

thermal and electrical networks through the capillary pores in the cement matrix.’

What is meant by larger here?

A: repeated phrases have been deleted.

Page 2, line 68 – not sure about the term ‘fracture direction elastic modulus’

A: The manuscript has been revised

transverse elastic modulus

Page 2, line 69

‘CNC can be classified as nanocellulose-like CNF’

A: The manuscript has been revised

CNC can be divided into CNF and nanocellulose

Page 4, line 140

The distributed CNC showed a rod shape of 40–100 nm or less3.  Provide scale bar in Fig. 2 a and Fig. 9

A: Fig. 2 and Fig. 9 have been revised

  1. Were the samples treated (e.g. sputter coating) before SEM analysis?

A: For SEM analysis, sputter coating was performed to improve the conductivity and image quality of the sample.

  1. Mention the ‘accumulated scans’ in IR measurement.

A: The manuscript has been revised

As for the peak position according to the accumulated scans of FT-IR is some peaks in the spectrum of the CNC that underwent the surface modification process using APTES showed a change (Fig. 3).

  1. Page 5, line 191 – give space between values and units

temperature increase of 20 °C/min from 0 to 1000 °C

A: The manuscript has been revised

temperature increase of 20 °C/min from 0 to 1000 °C

  1. As shown in Fig. 5, is the initiation of hydration region-specific or does it occur along

multiple points?

A: When CNC is distributed in the unhydrated area inside the cement, it improves the degree of hydration through the role of a water channel.

  1. Page 10, line 292, ‘…but no ductility effect was observed owing to the incorporation

of PCL (Fig. 7(b)). This is believed to be because of brittleness…’

A: The manuscript has been revised

but the ductility effect due to the PCL incorporation was not observed and showed a tendency to brittle fracture (Fig. 7(b)).

When/how the PCL transform from a ductile resin over to a brittle material.

A: In the case of CNC, it serves to increase hydration in the unhydrated area, and PCL (powder) represents a filler effect that fills the internal space of cement. As a result, the compressive strength was improved, which is judged to show a brittle tendency that appears at high strength.

Round 2

Reviewer 1 Report

The authors have revised the manuscript by responding point-by-point to my comments/suggestions. However, there are still point which the authors have overlooked:

- The Y-axis of the FTIR plots in Figure 3. It should be transmittance, not absorbance. 

- Standard deviation and error bars should be added for the data in Table(s) and graph (Figure 4), respectively.

Author Response

he authors have revised the manuscript by responding point-by-point to my comments/suggestions. However, there are still point which the authors have overlooked:

- The Y-axis of the FTIR plots in Figure 3. It should be transmittance, not absorbance. 

- Standard deviation and error bars should be added for the data in Table(s) and graph (Figure 4), respectively.

A: Graph revisions and standard deviations were added to Table 6.
